# Liquid Biopsy with Detection of NRAS^Q61K^ Mutation in Cerebrospinal Fluid: An Alternative Tool for the Diagnosis of Primary Pediatric Leptomeningeal Melanoma

**DOI:** 10.3390/diagnostics12071609

**Published:** 2022-07-01

**Authors:** Angela Mastronuzzi, Francesco Fabozzi, Martina Rinelli, Rita De Vito, Emanuele Agolini, Giovanna Stefania Colafati, Antonella Cacchione, Andrea Carai, Maria Antonietta De Ioris

**Affiliations:** 1Department of Hematology/Oncology, Cell and Gene Therapy, Bambino Gesù Children’s Hospital, IRCCS, 00165 Rome, Italy; angela.mastronuzzi@opbg.net (A.M.); francesco.fabozzi@opbg.net (F.F.); antonella.cacchione@opbg.net (A.C.); 2Laboratory of Medical Genetics, Bambino Gesù Children’s Hospital, IRCCS, 00165 Rome, Italy; martina.rinelli@opbg.net (M.R.); emanuele.agolini@opbg.net (E.A.); 3Department of Pathology, Bambino Gesù Children’s Hospital, IRCCS, 00165 Rome, Italy; rita.devito@opbg.net; 4Imaging Department, Bambino Gesù Children’s Hospital, IRCCS, 00165 Rome, Italy; gstefania.colafati@opbg.net; 5Department of Neurological and Psychiatric Sciences, Bambino Gesù Children’s Hospital, IRCCS, 00165 Rome, Italy; andrea.carai@opbg.net

**Keywords:** leptomeningeal melanoma, children, liquid biopsy, NRAS^Q61K^

## Abstract

Primary leptomeningeal melanoma (PLMM) is a very rare disease in childhood with a poor prognosis. NRAS^Q16K^ mutation frequently drives malignant transformation in this population, so its evaluation should be considered in childhood PLMM diagnosis. In the presented case, the mutation was detected by Sanger sequencing performed on DNA extracted from cerebrospinal fluid neoplastic cells. Liquid biopsy has been shown to be a safe and reliable technique for the diagnosis of PLMM. Its use can potentially be extended to other neoplasms of the central nervous system bearing well-defined molecular mutations, sparing the patient invasive surgery and finally allowing a more rapid diagnosis and early initiation of targeted therapies.

## 1. Introduction

Primary leptomeningeal melanoma (PLMM) has been reported in a few hundred patients with a peak of incidence in the fourth decade of life and unfavorable outcome. Pediatric experience is extremely limited since PLMMs account for about 0.1% of central nervous system (CNS) tumors. Leptomeningeal melanocytes are more susceptible to NRAS-driven transformation than their cutaneous counterpart. Moreover, characteristic molecular differences can be found between pediatric and adult PLMM: in fact, as we reported previously and according to a well-developed murine model by Pendersen et al., the mutated NRAS (NRAS^Q61K^) is strictly associated with childhood melanoma of the CNS, while GNAQ and GNA11 mutations drive adult melanoma [1,2]. Evaluation of NRAS^Q16K^ mutation should be considered in childhood PLMM diagnosis. Nevertheless, PLMM biopsy is an invasive procedure in the context of a disease with a dismal prognosis needing a prompt therapeutic strategy.

## 2. Materials and Methods

A PLMM was diagnosed in a 14-year-old girl referred to the Department of Hematology/Oncology, Cell and Gene Therapy at the Bambino Gesù Children’s Hospital. The clinical records, imaging, and pathology findings were reviewed for this report. All investigations were conducted according to principles expressed in the Declaration of Helsinki.

## 3. Results

A 14-year-old girl presented to our hospital with headache, vomiting, and unilateral VI nerve palsy. The magnetic resonance imaging (MRI) detected a diffuse leptomeningeal enhancement of the right frontal lobe and spine (Figure 1). Cytological examination of the cerebrospinal fluid (CSF) showed the presence of neoplastic cells with abundant pigmented cytoplasm, large vesicular nuclei, and prominent nucleoli suspected to be melanoma cells (Figure 2 and Figure 3). The immunocytochemical positivity for HMB45 and Mart1 confirmed the hypothesis. A right frontal craniotomy with *en-bloc* cerebral and leptomeningeal biopsy confirmed melanomatosis. The NRAS^Q61K^ mutation was detected by Sanger sequencing performed on DNA extracted from both tumor sample and CSF neoplastic cells. NRAS^Q61K^ is a hotspot mutation located in the GTP-binding region of the NRAS protein, conferring a loss of function as indicated by activation of downstream pathway signaling, increased survival, and transformation of cultured cells (Figure 4). Specifically, activated mutant NRAS^Q61K^ has been demonstrated to drive aberrant melanocyte signaling, survival, and invasiveness via Rac1-dependent mechanism [3]. Both CSF and tumor biopsy cells shared similar pathology and molecular aspects. Extra CNS localization was excluded and a diagnosis of PLMM in the absence of a neurocutaneous melanosis (NCM) was made. The patient was treated with conventional chemotherapy based on temozolomide and intrathecal liposomal cytarabine. A concomitant treatment with everolimus was administered in order to block NRAS activation by m-TOR inhibitor cascade, followed by ipilimumab to induce cytotoxic lymphocytes activation and expansion, plus nivolumab to inhibit PDL1. The disease was under control for six months, followed by rapid progression and death.

## 4. Discussion

Primary leptomeningeal melanoma is a rare cancer. The clinical manifestation and radiological changes are not specific; the diagnosis may be difficult with delay and misdiagnosed. Moreover, the treatment strategies are a challenge and the best option remains to be elucidated. The confirmed diagnosis is established by leptomeningeal biopsy or surgical tissue. The prognosis is dismal due to the inefficiency of chemotherapy and/or radiotherapy [1,2,4,5].

Current molecular profiling is becoming part of tumor classification with an impressive impact on diagnosis, prognosis, and ability to predict the response to treatment. The liquid biopsy with evaluation of the ‘liquid biome’ seems to be a promising option in CNS tumors. The identification of genetic alterations in cancers is crucial for precision medicine. However, surgical approaches to obtain brain tumor tissue should be invasive. Profiling circulating-tumor DNA in liquid biopsies has emerged as a promising approach to avoid invasive procedures and useful for diagnosis and to evaluate response to treatment. Clearly, a specific genetic alteration should be associated with a specific histology [6,7,8,9,10]. NRAS^Q16K^ mutation is a specific molecular alteration observed in childhood PLMM diagnosis.

Indeed, a liquid biopsy for the NRAS detection should be considered in patients with leptomeningeal involvement in order to achieve a safe diagnosis with a faster and less aggressive approach, especially in a tumor with a well-known molecular profile. Moreover, this allows an early start of targeted therapy wherever it is possible. Last but not least, the time needed for recovery from surgery can be avoided as the risks of neurological sequelae related to more aggressive surgery.

Clearly, the liquid biopsy on CSF presents some limitations due to the fluctuations and may not be able to meet clinical requirements in few case of minimal involvement. In the presented case, the diagnosis was performed with an amount of less than 4 mL; 1–1.5 mL were used for cytology, 1 mL for biochemistry, and 1 mL to detect NRASq61k mutation. There is no standardized amount of CSF for liquid biopsy in children and age should always be considered in order to modulate the amount of fluid. Prospective studies will clarify indications and procedure for liquid biopsy on CSF. On the other hand, the sensitivity and specificity of Sanger sequencing in detecting H3K27M mutation in diffuse midline gliomas on CSF-derived cfDNA is, respectively, 87.5% and 100%. The histone 3 allele-specific PCR and single gene Sanger sequencing assays have been developed to support the diagnosis of H3K27M-mutant diffuse midline gliomas, with 87.5% clinical sensitivity for CSF cfDNA when compared to tissue testing [11]. Further studies are needed to clarify the sensitivity and specificity of Sanger sequencing in detecting brain tumor cells in pediatric population.

Our experience confirms the use of the CSF liquid biopsy as a non-invasive tool for diagnosis of childhood PLMM. This innovative technique can be potentially extended to other CNS neoplasms with well-defined molecular mutations, especially in histologies where surgical treatment is not a cornerstone, opening new horizons for minimally invasive diagnosis and early initiation of target therapy. Furthermore, our case adds additional information to the limited amount of literature that confirms the presence of the NRAS^Q61K^ mutation in this rare childhood cancer: to our knowledge, this is the fourth case of PLMM without a neurocutaneous melanocytosis [2,4,5,12,13].

The liquid biopsy seems to represent a promising option in CNS tumors considering both primary and secondary malignancies. Unlike extracranial tumors, the biopsy of intracranial lesions is more invasive and sometimes biased because of neoplasm heterogeneity. Moreover, some CNS tumors are located in crucial regions (brain stem, thalamus, and spinal cord) with challenging biopsy. Early detection of primary tumor or CNS metastasis or relapse could have a major impact on prognosis. The tumor molecular profile with possible actionable mutations is almost heterogeneous and evolves over time. Any effort should be addressed in order to identify early CNS metastasis or relapses and to provide tailored therapy options. As a result, there is an urgent need to identify few reliable tumor biomarkers that could be used for diagnosis and prognosis [14,15,16].

The liquid biopsy should be considered in different cancers with a potential role in early detection of CNS relapses/metastasis and at the same time in identification of actionable mutations with a clinical impact.

## Figures and Tables

**Figure 1 diagnostics-12-01609-f001:**
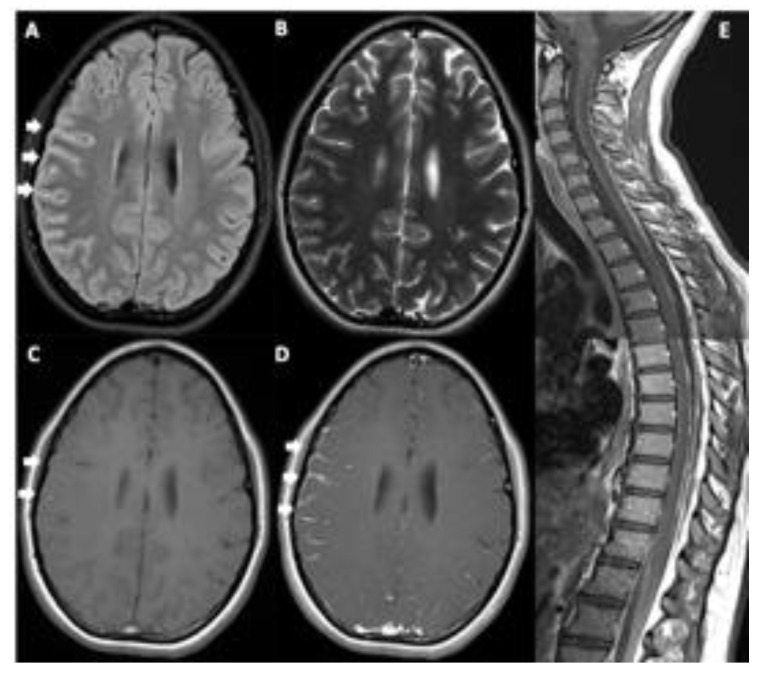
Brain MRI ((**A**–**D**), axial images) and spine ((**E**), sagittal Gd T1w image). Brain MRI shows diffuse leptomeningeal hyperintensity in the FLAIR image (arrows, (**A**)), not appreciable in the T2w image (**B**), with discontinuous mild spontaneous T1w hyperintensity (arrows, (**C**)) and diffuse leptomeningeal contrast-enhancement (Gd T1w, arrows, (**D**)). It is also appreciable diffuse spinal leptomeningeal contrast-enhancement (**E**).

**Figure 2 diagnostics-12-01609-f002:**
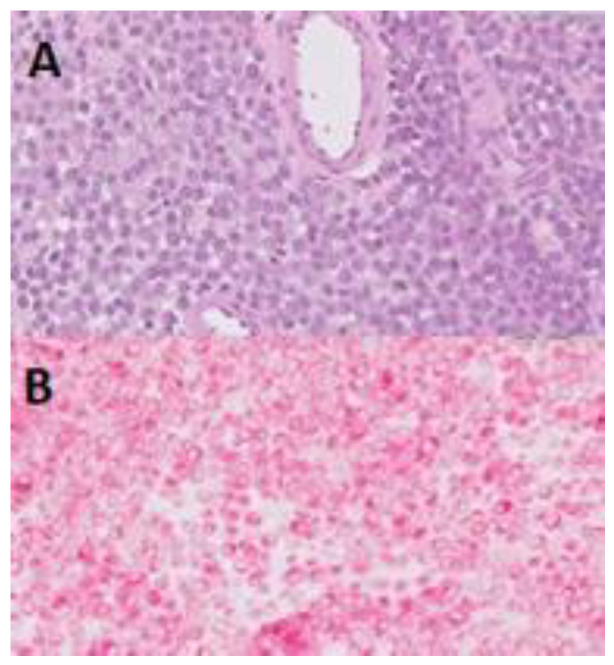
Histological examination of tumor biopsy. Neoplasm is composed of pleomorphic cells with vescicular nuclei, eosinophilic nuclear pseudoinclusion, and moderate cytoplasm (**A**,**B**).

**Figure 3 diagnostics-12-01609-f003:**
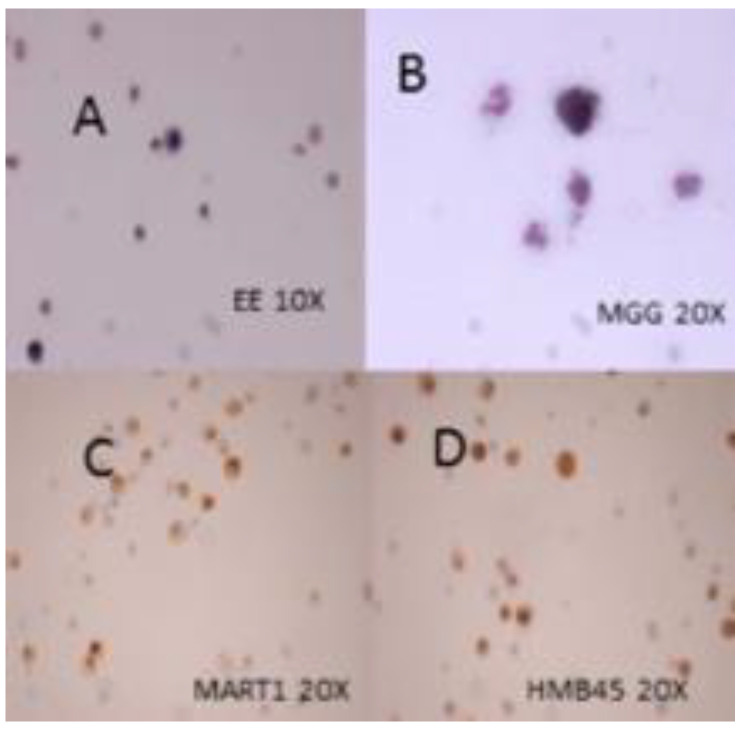
At cytological examination of the cerebrospinal fluid (CSF) stained with hematoxylin-eosin (**A**) and May–Grunwald–Giemsa (**B**) neoplastic cells with abundant pigmented cytoplasm, large vesicular nuclei and prominent nucleoli can be noted; immunocytochemical positivity for Mart1 (**C**) and HMB45 (**D**) confirms the hypothesis of melanoma.

**Figure 4 diagnostics-12-01609-f004:**
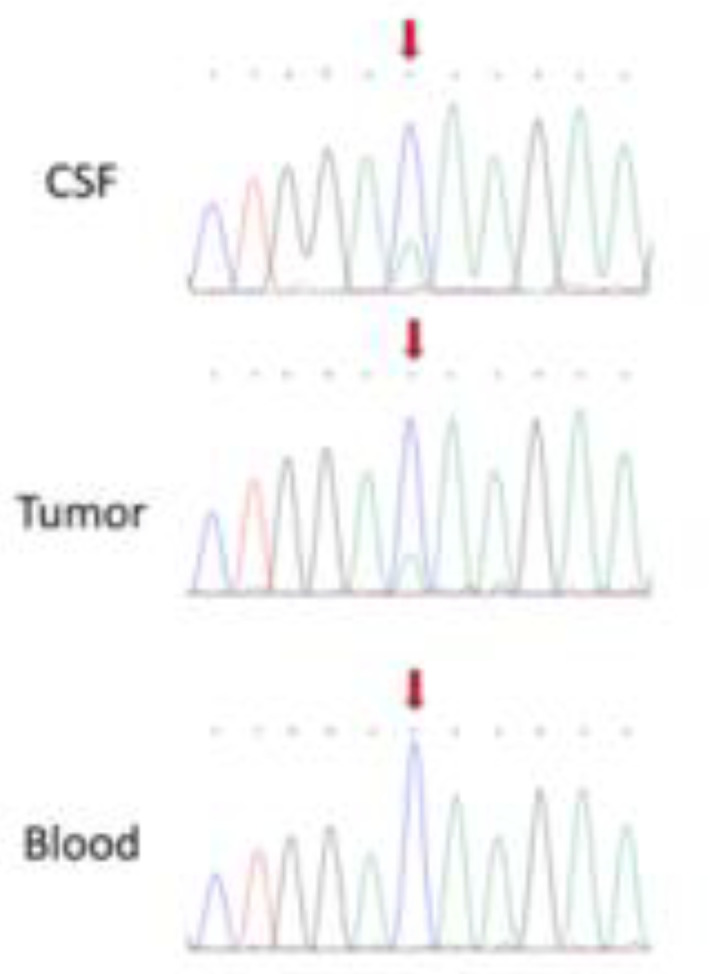
NRAS^Q61K^ mutation was detected by Sanger sequencing performed on DNA extracted from both tumor sample and CSF neoplastic cells. NRAS^Q61K^ is a hotspot mutation located in the GTP-binding region of the NRAS protein, conferring a loss of function as indicated by activation of downstream pathway signaling, increased survival, and transformation of cultured cells.

## Data Availability

Not applicable.

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
