# Peer review of "Liquid Biopsy with Detection of NRASQ61K Mutation in Cerebrospinal Fluid: An Alternative Tool for the Diagnosis of Primary Pediatric Leptomeningeal Melanoma"

_diagnostics, 2022, doi:10.3390/diagnostics12071609_

Round 1

Reviewer 1 Report

Reviewer's report

Manuscript ID: Diagnostics 1598581
Title:    Liquid biopsy with detection of NRASQ61K mutation in cerebrospinal fluid: an alternative tool for the diagnosis of primary pediatric leptomeningeal melanoma

Date: 2022/4/9

 Reviewer's report:
This is an interesting manuscript as it was a single case study of a highly rare  disease - primary pediatric leptomeningeal melanoma

The MS is well prepared, the major limitation is a single case report, lacking a strong evidence on the role of liquid biopsy for NRAS mutation finding. Nevertheless, its a novel study  which provide us a new perspective toward this rare disease.  Therefore, I'm sure this manuscript will add to a growing body of literature in the treatment  and evaluation of  this rare pediatric tumor. However, there's a few issue need to be answer  prior publication.

  1. What was the most characteristic image feature of PPLM on MRI ? Please show some important image study of  this disease on the  .
  2. How much CSF fluid samples is needed in liquid biopsy inoder to have a definite diagnosis ?
  3. The detection rates and specificity of liquid biopsy assays show high fluctuations and may not be able to meet clinical requirements. How do you overcome this problem ?
  4. What was the sensitivity and specificityof Sanger sequencing in detecting brain tumor cell ?

Author Response

Dear Editor,

We would like to thank for the precious revision.

We modified the text as suggested and answered the questions.

We add a point by point response to reviewers

Thank you for your kind attention

Maria Antonietta De Ioris,MD

Point by point

  1. What was the most characteristic image feature of PPLM on MRI ? Please show some important image study of  this disease on the  draft.

We added  representative images for PPLM features on MRI with few comments (SGC added to the author list)

2.How much CSF fluid samples is needed in liquid biopsy inoder to have a definite diagnosis ?

3.The detection rates and specificity of liquid biopsy assays show high fluctuations and may not be able to meet clinical requirements. How do you overcome this problem ?

4.What was the sensitivity and specificity of Sanger sequencing in detecting brain tumor cell ? 

Response question 2-3-4. The diagnosed was performed with an amount of less than 4 ml. We always use 1-1.5 ml for cytology, 1-1.5 ml for biochemistry  without any clinical problem. In this case 1 ml is used to detect NRASq61k mutation. Clearly there is no standardized amount of CSF for liquid biopsy in children and age should always be considered in order to modulate the amount of fluid. As observed there are fluctuations and probably in case of minor involvement should be difficult  to detect mutations. Prospective studies will clarify the limit of liquid biopsy performed on CSF.

The sensitivity and specificity of Sanger sequencing in detectingH3K27M mutation in diffuse midline gliomas on CSF-Derived cfDNA is  87.5% and 100%. (Histone 3 allele-specific PCR and single gene Sanger sequencing assays have been developed to aid the diagnosis of H3K27M-mutant diffuse midline gliomas, with 87.5% clinical sensitivity for CSF cfDNA when compared to tissue testing (Huang TY, Piunti A, Lulla RR, Qi J, Horbinski CM, Tomita T, James CD, Shilatifard A, Saratsis AM. Detection of Histone H3 mutations in cerebrospinal fluid-derived tumor DNA from children with diffuse midline glioma. Acta Neuropathol Commun. 2017 Apr 17;5(1):28. doi: 10.1186/s40478-017-0436-6. PMID: 28416018).  Further studies are needed to clarify the sensitivity and specificity of Sanger sequencing in detecting brain tumor cells in pediatric population. We thank RW1 who cleraly had focoused  the open questions of liquid biopsy on CSF

Reviewer 2 Report

Interesting paper looking at the clinical utility for liquid biopsy to detect NRASq61k mutation in CSF. 

Figure 1: s100 and Ki67 staining would be valuable

Figure 2: High power view of specific cells of interest would be helpful. 

Figure 3: data sufficient. 

The paper would benefit from expanded discussion regarding potential implications for other types of cancer: breast PMID: 32613208, lung PMID: 27340229. How can these findings dictate overall treatment strategies? A more comprehensive approach should be addressed PMID: 29604436

If the above is addressed, paper could be of interest to the readership.  

Author Response

Dear Editor,

We would like to thank for the precious revision.

We modified the text as suggested and answered the questions.

We add a point by point response to reviewers

Thank you for your kind attention

Maria Antonietta De Ioris,MD

Point by point

RW2

Interesting paper looking at the clinical utility for liquid biopsy to detect NRASq61k mutation in CSF. 

Figure 1: s100 and Ki67 staining would be valuable.

Unfortunately we have a limited amount of CSF so we omitted to perform Ki67 and we prefer to perform Mart1 that is more specific than S100

Figure 2: High power view of specific cells of interest would be helpful. 

We modified image resolutions

Figure 3: data sufficient. 

The paper would benefit from expanded discussion regarding potential implications for other types of cancer: breast PMID: 32613208, lung PMID: 27340229. How can these findings dictate overall treatment strategies? A more comprehensive approach should be addressed PMID: 29604436  

CSF liquid biopsy could represent an effective tools for early diagnosis of CNS relapse and metastasis with a therapeutic and prognostic impact. We added suggested papers, in order to underline potential implication for other types of cancer

If the above is addressed, paper could be of interest to the readership.

Round 2

Reviewer 2 Report

Authors failed to incorporate the references and address the key pathologies suggested. 

Please take another shot at revisions. Paper should be rejected if not addressed. 

Author Response

Dear Editor, Dear Reviewer

CSF liquid biopsy could represent an effective tools for early diagnosis of CNS relapse and metastasis with a therapeutic and prognostic impact. We added suggested papers, in order to underline potential implication for other types of cancer. We add the correct file version with the suggested papers in the reference list.  MA

Round 3

Reviewer 2 Report

Accept